# Capturing Extreme Events in Turbulence using an Extreme Variational Autoencoder

## Abstract

Turbulent flows are characterized by intense generation of turbulent kinetic energy through nonlinear physical processes which cascade from the large- to small-scale structures in a forward energy cascade, which is chaotic in nature, and statistically intermittent. Using a recently developed extreme variational autoencoder (XVAE), the turbulent flow fields are replicated to a high order of accuracy. In this extended abstract, we demonstrate XVAE as a powerful alternative to the classical Proper Orthogonal Decomposition (POD) technique for reconstructing large-eddy-simulation (LES) data for scalar temperatures from a buoyant turbulent field at a high Reynolds number of $10^{10}$.

*Keywords:* Turbulence, Spatial extremes, POD, Variational autoencoder

## 1   Introduction

Simulating turbulent flows numerically becomes computationally prohibitive as the Reynolds number increases. Proper Orthogonal Decomposition (POD) has been the most successful statistical method, extracting dominant energetic structures from turbulent flow fields as eigenfunctions of the two-point correlation tensor [Berkooz et al., 1993]. However, POD neglects the intermittency and also extreme turbulence events in the flow. This limitation leads to inaccuracies and information loss in the emulation process.

Variational autoencoders [VAEs; Kingma and Welling, 2013] offer an alternative, encoding data inputs as latent variables for generating new output realizations. However, traditional VAEs use Gaussian distributions and do not naturally account for extremes. We introduce a novel framework that incorporates dependent extremes within a VAE engine, allowing for flexible modeling of extremal dependence and efficient uncertainty quantification (UQ). Specifically, we develop a flexible max-infinitely divisible (max-id) process embedded in the VAE's structure (XVAE), enabling a robust emulation of complex turbulent flow systems [Zhang et al., 2023].

In this study, we apply both POD and XVAE to emulate turbulence flow from a high-resolution large-eddy simulation (LES) at the $x$-$z$ plane of a fixed $y$ level. We also extend the XVAE to allow for the emulation of 3D indexed data so we can model and emulate the evolution of turbulent plumes in all three different directions. We evaluate the methods based on their ability to emulate full data and joint tail behavior, with XVAE showing particular strength in UQ for extreme events.

## 2   Large-Eddy-Simulation of a Turbulent Plume

A modified version of the Advanced Research Weather Research and Forecast Model (WRF-ARW v4.1) [hereafter WRF; Skamarock et al., 2008] was used to simulate a turbulent buoyant plume

Submitted to Workshop on Bayesian Decision-making and Uncertainty, 38th Conference on Neural Information Processing Systems (BDU at NeurIPS 2024). Do not distribute.

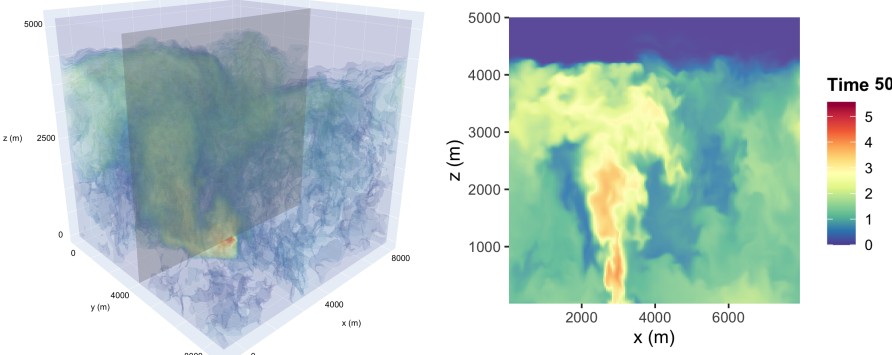

Figure 1: The left panel shows the log scale of marginally-transformed LES simulation of the Mosquito wildland fire at the 50th time; see Equation (1). The right panel shows the centerline data in the $x$-$z$ plane at $y = 4$ km. See Figure 2 for the corresponding emulation results.

via LES with active scalar transport equations, validated for buoyant plumes [Bhaganagar and Bhimireddy, 2020, Bhimireddy and Bhaganagar, 2021].

The simulation involved a heated, axisymmetric source with diameter $D$ in a quiescent environment. The LES model solves the compressible Euler equations in flux formulation, conserving mass, momentum, and energy; see Bhaganagar and Bhimireddy [2020] for details of the LES methodology and the validation. In this study, the LES coupled with WRF is used to simulate the Mosquito wildland fire (California, September 2022) using realistic boundary conditions from NOAA's HRRR model. The plume is released from a 400m diameter circular source at ($39.006°$ N, $120.745°$ W). The background atmospheric velocity profile, temperature profile and surface heat flux, were obtained from the WRF-LES simulations initiated at 11am on September 09.

The LES domain measured 8 km $\times$ 8 km $\times$ 5 km with a uniform Cartesian grid ($198 \times 198 \times 500$ nodes). Periodic boundary conditions were set on the sides, and constant-pressure at the top. At the center of the bottom boundary, the plume source had a prescribed buoyancy flux of $1.0710^4 m^4 s^3$, with no initial momentum, driven purely by buoyancy. Simulations ran for 50 minutes with data recorded every 30 seconds (i.e., 100 times in total). Planar data were extracted from the $x$-$z$ plane at $y = 99$ grid location (i.e., 4 km), the centerline of the plume.

## 3   VAE with spatial extremes

The turbulent buoyant plume exhibits high spatial irregularity, temporal volatility, and significant spatial clustering of extremes with anisotropy. Accurate spatio-temporal modeling is challenging due to its high-resolution grid. Therefore, we need to carefully model the tail of the joint distribution while ensuring computational efficiency. Zhang et al. [2023] extended the max-id process, and developed a model to allow for both short-range asymptotic dependence along with mid-range asymptotic independence, and long-range exact independence. They were able to emulate the extreme values in Red Sea sea surface temperature in very high dimensions for the first time.

Denote the observed turbulent plume at $t$ by $\{u_t(\boldsymbol{s}) : \boldsymbol{s} \in \mathcal{S}\}$, in which $\mathcal{S}$ may either be the 3D domain of 8 km $\times$ 8 km $\times$ 5 km or the 2D centerline; see Figure 1. In our study, we marginally transform the data so the input data conform to the max-id model assumptions

$$X_t(\boldsymbol{s}) = 1 - u_t(\boldsymbol{s}) \times 10^5. \tag{1}$$

Then $\{X_t(\boldsymbol{s}) : \boldsymbol{s} \in \mathcal{S}, \ t = 1, \ldots, 100\}$ are modeled as follows:

$$X_t(\boldsymbol{s}) = \epsilon_t(\boldsymbol{s}) Y_t(\boldsymbol{s}), \ \boldsymbol{s} \in \mathcal{S}, \tag{2}$$

where $\epsilon_t(\boldsymbol{s})$ is a noise process with independent Fréchet$(\mu, \tau, 1/\alpha_0)$ marginal distributions, where $x > 0$, $\tau > 0$ and $\alpha_0 > 0$. Then, $Y_t(\boldsymbol{s})$ is constructed using a low-rank representation:

$$Y_t(\boldsymbol{s}) = \left\{ \sum_{k=1}^{K} \omega_k(\boldsymbol{s}) Z_{kt} \right\}^{\alpha_0}, \tag{3}$$

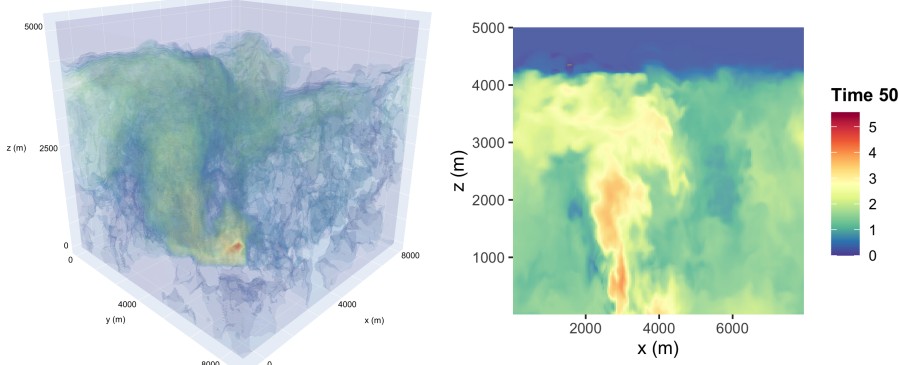

Figure 2: The left panel shows the emulation of the marginally-transformed LES simulation using the 3D XVAE and the right panel shows the emulation of cross section of $y = 4$ km using the 2D XVAE. See Figure 1 for the original turbulent plume input.

where $\alpha \in (0,1)$, $\{\omega_k(s) : s \in \mathcal{S}, k = 1, \ldots, K\}$ are a set of basis functions, and $\{Z_{kt} : k = 1, \ldots, K\}$ are independent exponentially-tilted positive-stable (expPS) variables, controlled by a rate parameter $\alpha_t \in (0,1)$ and a tilting parameter $\theta_{kt} \geq 0$. See Section B.1 in Zhang et al. [2023] for more specific forms of an expPS$(\alpha, \theta_k)$ distribution.

The hierarchical construct of our model (2)–(3) allows the embedding of the latent variables $Z_{kt}$, $k = 1, \ldots, K$ in encoded space of a VAE so we can encode and decode the turbulent plume fields while minimizing reconstruction error. The encoder and decoder are multilayer perceptron (MLP) neural networks that optimize parameters to retain maximum information and replicate the joint tail properties of the original field (i.e., to minimize the evidence lower bound). We refer to this VAE, integrated with the max-id process, as XVAE.

Once the XVAE is trained and its weights and biases are optimized, posterior simulation of new latent variables $\mathbf{Z}_t$ for the observations $\mathbf{X}_t$ at time $t$ can be performed efficiently. Synthetic data $\mathbf{X}_t^*$ can then be generated rapidly by passing these latent variables through the decoder and sampling from the model specified by (2)–(3). To transform $\mathbf{X}_t^*$ back to the original scale of the turbulence plume, we apply an inverse marginal transformation: $\mathbf{u}_t^* = (1 - \mathbf{X}_t^*)/10^5$, yielding the final emulated output. Refer to Zhang et al. [2023] for an explicit architecture of the XVAE.

## 4   Results

For both the 2D and 3D emulation tasks, we pre-specify a dense grid of $K$ knots $\{s_1^0, \ldots, s_K^0\}$ in the domain $\mathcal{S} \in \mathbb{R}^2$ or $\mathbb{R}^3$. The $k$th basis function is the Gaussian radial basis function, i.e., $\omega_k(s) \propto \exp\{-||s - s_k^0||^2/c\}$ for $s \in \mathcal{S}$, $k = 1, \ldots, K$. We follow Algorithm 1 in Zhang et al. [2023] to choose the best number of knots $K$ and shape parameter $c$,

Figure 2 displays emulation results at time 50 using the XVAE for both 2D and 3D LES. Generally, XVAE closely replicate the spatial patterns of turbulence across all times. More importantly, multiple model evaluation metrics, such as the tail RMSE and the spatially-pooled QQ-plot, show good performance in fitting and emulating both the full range of data and the joint tail behavior.

In addition, we conduct a comparative study in the 2D case to evaluate the emulation performance of XVAE against POD, commonly used to capture dominant structures in 2D turbulent flows. We compute an extremal dependence measure from spatial extremes literature:

$$\chi_{ij}(p) = \Pr\{F_j(u_j) > p \mid F_i(u_i) > p\} = \frac{\Pr\{F_j(u_j) > p, F_i(u_i) > p\}}{\Pr\{F_i(u_i) > p\}} \in [0,1], \qquad (4)$$

in which $p \in (0,1)$ and $F_i$ and $F_j$ are the marginal distribution functions for $u_i$ and $u_j$, respectively. Assuming stationarity and isotropy, we simplify $\chi_{ij}(u)$ to $\chi_d(u)$, with $d = ||s_i - s_j||$ denoting the distance. The metric $\chi_d(p)$ is then empirically estimated for each data set (observed or emulated).

Figure 3 compares empirical measure $\widehat{\chi}_d(p)$ for $p \in (0.9, 0.9999)$ at $d = 4.24$ and $d = 8.49$ for the observed turbulent buoyant plume and the emulations from POD (50 modes) and XVAE. The

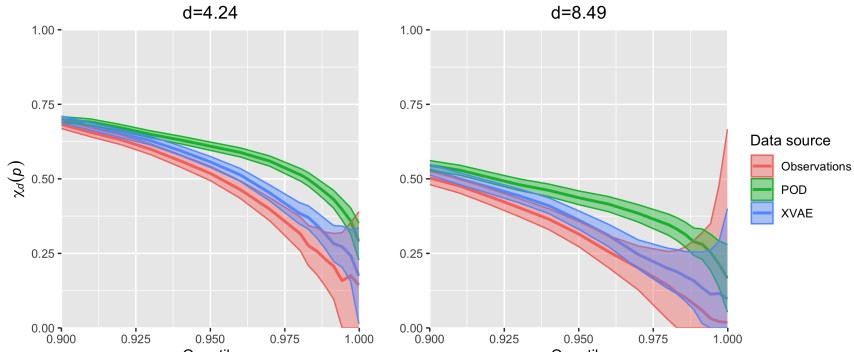

Figure 3: The empirically-estimated $\chi_d(p)$ at $d = 4.24$ (left) and $d = 8.49$ (right) based on plume observations (red), POD reconstructed data using 50 modes (green) and XVAE emulated data (blue).

XVAE aligns closely with observed data at both distances, while POD tends to overestimate extremal dependence at sub-asymptotic levels ($p \in (0.93, 0.99)$), likely due to oversmoothing.

To investigate how the marginal distributions at different locations evolve over time, we generate 1,000 emulated samples of $\{\mathbf{X}_t : t = 1, \ldots, n_t\}$ using XVAE. For each location and time point, we thus obtain 1,000 replicates to assess variational Bayes-based marginal uncertainty. Figure 4 shows the marginal distributions of $\log(X_{it})$ for $i = 500, 3861, 17323, 80017, 83014$ at $t = 1$ and $100$. The plots reveal how the density shapes change over time at each location, offering sensible UQ for understanding the range and likelihood of turbulence intensity at specific locations and times.

## 5   Conclusion

The proposed XVAE method improves the emulation of turbulent buoyant plumes, especially in capturing joint extremal behavior. This approach has significant potential for enhancing predictive models of extreme events. Moreover, the XVAE allows for parameter estimation and UQ within a variational Bayesian framework. This helps in assessing how sensitive the LES is to changes in input parameters and in evaluating the probability of extreme clusters, such as those involving the transport of hazardous or contaminant gases in the atmosphere, essential for improved risk management and more robust decision-making.

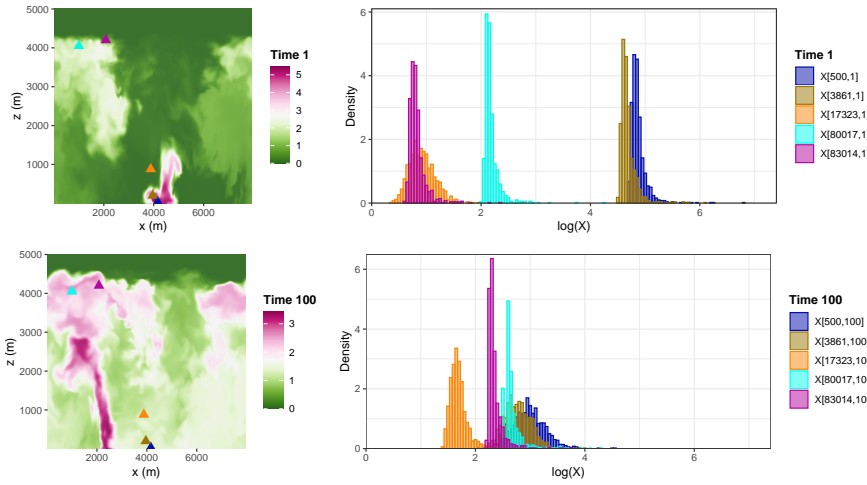

Figure 4: The left panels show the log scale of $\mathbf{X}_1$ and $\mathbf{X}_{100}$. The right panels show the histograms based on 1,000 samples from the emulation. We present density histograms individually for 5 specific sites (marked in the left panels) at the two times.

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
