# OpenReview forum: "Capturing Extreme Events in Turbulence using an Extreme Variational Autoencoder"
_NeurIPS.cc/2024/Workshop/BDU — NeurIPS BDU Workshop 2024 Poster_

### Official Review · Reviewer_2Nv8 · 2024-09-16
**Promising XVAE Method Shows Potential, but Requires Enhanced Comparative Analysis, Model Specifications, and Dataset Transparency**

**Rating:** 7
**Confidence:** 4

**Review:**

The paper demonstrates several commendable aspects:
1. The writing is clear and well-structured.
2. The proposed XVAE method is innovative and shows promise.
3. The visualizations are both effective and intuitive.
4. XVAE exhibits superior performance, particularly in capturing extreme events, when compared to the traditional Proper Orthogonal Decomposition (POD) technique.

Areas for Improvement
1. The paper would benefit from a more comprehensive comparative analysis. Include comparisons with recent state-of-the-art models to provide a more current context. The POD technique in 1993 is relatively outdated. Comparing XVAE with more contemporary methods would strengthen the paper's relevance and impact.
2. To enhance reproducibility and clarity, the paper should include the number of knots used in the best-performing model and the shape parameters for the optimal training result. These details are currently absent but are crucial for a thorough understanding of the model's architecture and performance.
3. The paper would be improved by providing a detailed description of the dataset used in the study, including the dataset as an attachment or providing a link to access it.

---

### Official Review · Reviewer_spGp · 2024-10-09
**A paper about using VAEs to forecast turbulent flows with a focus on the role of rare events**

**Rating:** 6
**Confidence:** 3

**Review:**

This submission proposes a VAE extension designed for modeling turbulent flows. An important part of this is sensibly handling non-Gaussianity, since correctly accounting for the role of rare events is important due to non-linear behavior affecting statistical properties of turbulent flows. For this, the authors study a purpose built VAE extension where Gaussianity in latent space is replaced with a significantly more complicated distribution designed to handle rare events.

I am not an expert on the physics / weather modeling side and cannot offer feedback beyond surface level on those aspects of the work. However, I am an expert on the ML side, and I think the the authors' focus on modeling tail behavior through a domain specific non-Gaussian extension of VAEs makes sense. This kind of model would not have been very explored historically in this domain due to the usual objections involving non-convexity of the resulting optimization problems, so it makes sense to explore it now as part of a workshop paper.

The main weakness of the work is its fit to the workshop. While VAEs can in principle be seen as a Bayesian approach, this connection is substantially more shaky than other topics listed in the call for papers. The flavor is much less Bayesian learning and uncertainty, and much more deep representation learning with a carefully constructed latent space. However, the workshop's list of topics includes "Forecasting for dynamical systems" which this paper one could argue is about.

On balance of the tradeoffs I vote to accept.

---

### Decision · Program_Chairs · 2024-10-09

Accept (Poster)